# Barriers, Enablers, and Perceptions on Dietary Salt Reduction in the Out-of-Home Sectors: A Scoping Review

**DOI:** 10.3390/ijerph18158099

**Published:** 2021-07-30

**Authors:** Viola Michael, Yee Xing You, Suzana Shahar, Zahara Abdul Manaf, Hasnah Haron, Siti Nurbaya Shahrir, Hazreen Abdul Majid, Yook Chin Chia, Mhairi Karen Brown, Feng J. He, Graham A. MacGregor

**Affiliations:** 1Dietetics Programme and Centre for Healthy Ageing and Wellness (H-Care), Faculty of Health Sciences, Universiti Kebangsaan Malaysia, Jalan Raja Muda Abdul Aziz, Kuala Lumpur 50300, Malaysia; violamichael@moh.com.my (V.M.); youyeexing@ukm.edu.my (Y.X.Y.); zaharamanaf@ukm.edu.my (Z.A.M.); 2Disease Control Division, Ministry of Health Malaysia, Putrajaya 62590, Malaysia; drsitinurbaya@gmail.com; 3Nutritional Sciences Programme and Centre for Healthy Ageing and Wellness (H-Care), Faculty of Health Sciences, Universiti Kebangsaan Malaysia, Jalan Raja Muda Abdul Aziz, Kuala Lumpur 50300, Malaysia; hasnaharon@ukm.edu.my; 4Centre for Population Health, Department of Social and Preventive Medicine, Faculty of Medicine, University of Malaya, Kuala Lumpur 50603, Malaysia; hazreen@ummc.edu.my; 5Department of Nutrition, Faculty of Public Health, Universitas Airlangga, Jawa Timur 60115, Indonesia; 6Department of Medical Sciences, School of Medical and Life Sciences, Sunway University, Petaling Jaya, Selangor Darul Ehsan 47500, Malaysia; ycchia@sunway.edu.my; 7Department of Primary Care Medicine, Faculty of Medicine, University of Malaya, Kuala Lumpur 50603, Malaysia; 8Wolfson Institute of Preventive Medicine, Barts and The London School of Medicine and Dentistry, Queen Mary University of London, London EC1M 6BQ, UK; mhairi.brown@qmul.ac.uk (M.K.B.); f.he@qmul.ac.uk (F.J.H.); g.macgregor@qmul.ac.uk (G.A.M.)

**Keywords:** barriers, enablers, perception, salt reduction, out-of-home

## Abstract

In this review, we have investigated the perceptions, barriers, and enabling factors that were responsible for a dietary salt reduction in the out-of-home sectors. For this purpose, we examined different knowledge databases such as Google Scholar, Ebscohost, MEDLINE (PubMed), Ovid, and Cochrane Library for research articles from September to December 2020. The inclusion criteria for the research articles were that they had to be published in English and had to be a qualitative or quantitative study that was conducted after 2010. These studies also had to report the various enablers, barriers, and perceptions regarding salt reduction in the out-of-home sectors. After implementing the inclusion criteria, we successfully screened a total of 440 studies, out of which 65 articles fulfilled all the criteria. The perceived barriers that hindered salt reduction among the out-of-home sectors included lack of menu and food variabilities, loss of sales due to salt reduction, lack of technical skills for implementing the salt reduction processes for cooking or reformulation, and an absence of environmental and systemic support for reducing the salt concentration. Furthermore, the enablers for salt reduction included the intervention programs, easy accessibility to salt substitutes, salt intake measurement, educational availability, and a gradual reduction in the salt levels. With regards to the behavior or perceptions, the effect of organizational and individual characteristics on their salt intake were reported. The majority of the people were not aware of their salt intake or the effect of salt on their health. These people also believed that low salt food was recognized as tasteless. In conclusion, the enablers, barriers, and perceptions regarding salt reduction in the out-of-home sectors were multifaceted. Therefore, for the implementation of the strategies, policies, and initiatives for addressing the barriers, the policymakers need to encourage a multisectoral collaboration for reducing the salt intake in the population.

## 1. Introduction

A majority of the countries, worldwide, have undertaken steps to decrease the salt intake in their population. Most of the salt reduction strategies such as legislation, reformulation, and salt reduction intervention programmes were suggested by the industries as well as governmental and non-governmental organizations [1,2]. Despite the probable health benefits of a low salt intake, very few researchers have investigated the implementation of the salt reduction intervention for achieving a sustained reduction of salt intake among the Low- and Middle-Income Countries (LMICs) [3,4]. It was noted that translational research could help in determining the available intervention steps which were effective and suitable for various contexts. In an earlier study, the researchers concluded that based on the moderate quality of the evidence, the population-level interventions could improve the salt-related behaviour of the people [5].

A systematic review conducted by Cappuccio et al. (2015) indicated that some of the successful policies included comprehensiveness, which involved health education, population monitoring, and reformulation for decreasing the salt concentration included in the processed foods (responsible for ≥75% of the daily salt intake) [6]. These population-based salt reduction policies were rapid, equitable, powerful, and enabled cost saving.

The United Kingdom developed an excellent and popular national salt reduction strategy in 2003. These policies included three major factors, i.e., improved nutrition labelling, consumer awareness campaigns, and a government-backed salt reduction model which set low salt targets for more than 85 different food categories that had to be achieved by the food manufacturers within a specified time frame [7]. This resulted in a 15% decrease in the salt intake within the population in the United Kingdom (from 9.5 to 8.1 g/day, in the period ranging from 2001 to 2011) [7]. Very few studies have shown that salt intake was influenced by income. Azizan et al. (2020) carried out an intervention study and noted that dietary modification could improve the nutritional intake such as the salt intake and the biochemical profile of the low-income urban population suffering from hypertension in Malaysia [8].

The out-of-home sector consists of the outlets which serve foods and drinks for immediate consumption or take away. This sector includes formal and informal sections. A formal out-of-home sector that includes the licensed and registered cafes, restaurants, school/work canteens, fast food outlets, takeaways, food delivery services or hotel restaurants. On the other hand, the informal out-of-home sector includes street hawkers, food vendors, and food delivery services [9].

Commonly, many people worldwide, are choosing to dine outside their homes [10,11]. A survey was carried out in the USA, which showed that ≥50% of the adults eat out more than 3 times per week, while 35% of the adults eat fast food meals at least 2 times every week [12]. In 2012, another study was conducted in China, where the researchers determined the relationship between the nutritional status of the residents and the onset of chronic diseases. The researchers noted that 35.5% of the residents, aged above 6 years preferred dining out. Out of these, 42.2% were urban residents, while 28.5% of the people were rural residents [13]. The home-cooked meals were regarded as healthier compared to those purchased from out-of-home places [14,15]. Furthermore, the out-of-home meals contained higher levels of salt, added sugar and saturated fats, while they were also low in vitamins, fiber, and minerals compared to the home-cooked meals [16]. The research question for this review paper is “What are the barriers, enablers and perceptions on dietary salt reduction in the out-of-home sectors?” Hence, this review was carried out for identifying the different barriers, motivators, and perceptions of the people regarding the reduction of dietary salt levels in out-of-home meals. This review would also help in identifying the different strategies that can be used for salt levels reduction.

## 2. Materials and Methods

In this study, the researchers have adopted the framework proposed by Arksey and O’Malley (2005) for scoping different reviews [17], along with the enhanced and adapted methodologies proposed by Levac et al. (2010) and Daudt et al. (2013) [18,19]. Based on all these frameworks, the researchers implemented five stages, such as (i) Identifying research questions; (ii) Determining important studies; (iii) Selection of the studies; (iv) Charting data; and (v) Collating, summarizing, and presenting all the results. This framework is the most commonly used framework in the scoping review. The researchers did not use ethical registration since this was a scoping review.

### 2.1. Identification of the Relevant Study

A search strategy was developed collectively. Regular discussions and meetings were conducted for determining the significance of all citations and resolving disagreements. The first author (V.M.) primarily conducted the search and two co-authors (Y.X.Y. and S.N.S.) independently searched using the same keywords. Consolidation of articles to be included in the review was done among these three authors. Some inclusion criteria were set before selecting the citations. These criteria stated that the study must be published in the English language, must be peer-reviewed or grey literature, and must be published between 1 January 2010 and 31 December 2020. These reports must also describe and highlight the enablers, barriers, and perceptions related to the sodium or salt reduction amongst the people, along with the steps undertaken by the policymakers or out-of-home sectors for implementing these steps. On the other hand, the studies that were not original editorials, book chapters, research articles, reviews, opinions, and commentaries or did not include new information and knowledge were not included in this review. Moreover, the studies that presented only summaries of earlier reports were not included in the review.

Perceived barriers are defined as a person’s estimation of the level of challenge of social, personal, environmental, and economic obstacles to a specified behavior or their desired goal status on that behavior [20]. While, enablers are defined as the one that enables another to achieve an end in salt reduction strategies [3].

The search strategy used in this study was a combination of various keywords such as (Barriers OR enablers OR perceptions OR facilitators) AND (salt OR sodium reduction)) AND (out-of-home OR street OR restaurants OR cafeteria OR work place OR food industries). The synonyms and keywords generated after using the search terms are presented in Table 1.

The search strategy was implemented on different databases such as Google Scholar, Ebscohost, MEDLINE (PubMed), Ovid, and Cochrane Library. After using the inclusion and exclusion criteria, the search yielded 440 studies, out of which 65 articles (three qualitative and 62 quantitative studies) fulfilled all the criteria. Figure 1 presents the flow chart diagram implemented in the study.

### 2.2. Extraction and Analysis

Table 2 presents a summary of the study characteristics included in the review. Thereafter, the researchers extracted and tabulated each abstract and full text of the final 65 studies included after the final analysis, for highlighting the year, survey, country, study design, participants, methodologies, results, and conclusions (Appendix A Appendix A). We analyzed every study for categorizing the studies into three major themes, i.e., barriers, enablers, and perceptions.

## 3. Results

Table 3 summarizes the findings related to barriers, enablers, and perceptions of dietary salt reduction in the out-of-home sectors.

### 3.1. Barriers of Dietary Salt Reduction in the Out-of-Home Sectors

The findings for the barriers which prevented salt reduction in the out-of-home sectors included the lack of menu and food variabilities, lack of technical skills for cooking, profit loss due to salt-reduced foods, lack of technical expertise for reformulation, and other factors as listed in Table 3.

### 3.2. Enablers of Dietary Salt Reduction in the Out-of-Home Sectors

Enablers are described as factors that help in the implementation of salt reduction programmes, strategies, initiatives, and interventions for various out-of-home sectors. Enabling factors that encouraged a salt reduction in the out-of-home sectors included the salt reduction intervention program, food analysis studies, salt substitute accessibility, gradual reduction of salt approach, training availability, and other factors which were categorized separately as listed in Table 3.

### 3.3. Perceptions of Dietary Salt Intake in the Out-of-Home Sectors

The perceptions of people on salt intake in out-of-home sectors are shown in Table 3. These included the negative effects of a high salt diet, low salt foods recognized as tasteless, and individual or organizational characteristics effects in salt reduction.

## 4. Discussion

The out-of-home sectors that have been considered in this review include catering operators, food industries, school children, workplace organizations, and food industries. Food industries worked hand-in-hand with the policymakers and implemented interventions for decreasing the total salt content. The majority of the interventions which targeted the foodservice settings and work places were implemented in the USA, South Korea, and China [21,31,63].

### 4.1. Dietary Salt Intake and Sources

The studies carried out in the USA showed that salt was added to the diet from different packaged/non-packaged foods or foods sourced from out-of-home settings. This was responsible for a total of 67–70% of the salt intake among the population [64,65]. Similarly, studies conducted in the European Union indicated that the people had a higher salt intake, wherein 75–80% of the salt intake was due to the consumption of processed foods, 10–15% of the salt was added during the cooking process or at the table, while 5–10% of the salt was naturally present in different food items. However, in developing and LMIC countries, the salt that was used for seasoning played a vital role [29,83]. A recent study showed that the intake of salt was increasing amongst the Chinese population [78]. A majority of the processed foods that were available in the Chinese local supermarkets were high in salt content [84].

This scoping review noted that a majority of the countries such as the UK, USA, Australia, European countries, South Korea, and Brazil have set a few national voluntary targets [45,48]. This was implemented after many reformulation efforts were carried out in the food products such as meat, dairy products, bread, and soups [29,46,58]. This was considered a very effective intervention as it helped in averting 0.7–1.9 million deaths in 10 years using a modelling approach, as proposed by Coxson et al. (2013) [82]. Moreover, mandatory salt reduction targets using legislation were achieved in South Africa for many food products [28].

### 4.2. Barriers for Dietary Salt Reduction

A lack of availability of low salt foods, lesser food choices or menu variability and incomprehensive salt data were regarded as major barriers that affected the out-of-home sectors in this review. These issues were attributed to a lack of technical skills for reformulation and cooking low salt foods [26]. An absence of food analysis techniques for measuring the salt concentration in the foods was also an issue.

Issues related to the lack of technical skills in preparing or producing low salt foods have to be resolved as they can lead to food safety problems, concerns regarding consumer acceptance, higher costs, and complications arising due to the use of various sodium alternatives. This can be a factor that could demotivate the food manufacturers in European countries [29]. In addition, several environmental factors such as pressure for maintaining profit margins and fear of decreased sales from the industrial sectors act as barriers that prevent the production of low salt foods [21,22,47]. Many of the barriers listed in this scoping review are dependent on capacity building and can easily be eliminated through proper training, intervention or education as highlighted in the Enablers findings.

### 4.3. Enablers for Dietary Salt Reduction

The effective intervention strategies include product substitutions [25], recipe modification [40], social marketing approach [39], comparing the salt concentration in different pre-packaged foods based on the available data such as sales-weighted sodium concentration and sodium data presented on the Nutrition Facts panel provided in the restaurants and point of purchase, offering low salt foods and encouraging the consumers to buy low sodium food products.

Salt substitutes that are used for reformulating the food products such as potassium chloride can be used for replacing the sodium chloride salts in various products [33,79]. Additionally, potassium citrate can be used for replacing salt in different types of bread [34] or herbs can be used as a salt substitute while cooking foods at home [33]. Salt substitutes such as potassium chloride could reduce dietary sodium intake and cause an increase in dietary potassium results in the decreased incidence of hypertension [79]. Reformulating food products to lower sodium levels using potassium chloride may actually result in positive health effects in the general population [79]. The limited availability of different low salt food products in the supermarkets, cafeteria, work-places, vending machines, and kiosks is attributed to the inability of the food manufacturing companies in reformulating their food products [48] or a lack of technical skills by the workers and chefs. Hence, the policymakers need to develop policies to reformulate the foods using various salt substitutes [41] and provide technical training to the different personnel with regards to food preparation, food procurement or marketing [63]. In one wholesale survey study, the researchers noted that when people were offered nine types of low salt bread that were prepared using potassium citrate, they accepted the products and did not voice any complaints regarding the flavor changes. They further stated that the respondents were unaware of salt substitution and reduction [34]. This study indicated that it is possible to reduce the salt concentration in foods using appropriate salt substitutes, without altering the taste of the food.

For ensuring salt reduction, the policymakers have to carry out a food analysis study and update their databases to provide information related to the salt content in the packed foods or food acquired from out-of-home sectors to monitor and compare the salt targets and provide better food labels [74]. The salt concentrations can be measured after a 24 h urine analysis along with the blood pressure after 6 months of three educational interventions among 155 of 10–12 years old school children. This helps in determining if the salt reduction intervention is effective or not [33]. It is important to identify the partners having a shared experience and common objectives and who aim for a sustainable salt reduction [40].

Effective legislation is one of the salt reduction strategies that has been implemented in many countries such as South Africa, Argentina, Belgium, Bulgaria, Netherlands, Portugal, and Paraguay [23,75,85,86,87]. For instance, South Africa and Argentina have set the maximum total sodium levels for 13 and 18 categories of food products, respectively that started in 2013 [85,88]. The other countries have also implemented the legislation on salt reduction through the regulation by setting the maximum level of sodium content in food products such as breads, cheese, meat, and poultry [75,86,87].

Appropriate training must be provided to the staff for food preparation, food procurement, and marketing [63]. It helped the employees improve their skills and learn new techniques for preparing low salt food such as in the Korean catering firms [21]. The staff should also be trained to improve their knowledge for reducing salt intake [43].

Other enabling factors which helped in reducing the salt intake included the maintenance of customer demands, implementation of effective strategies, and support. Monitoring and establishing a partnership between the state, local, and national health organizations is necessary [48]. The psychological characteristics of the stakeholders and environmental factors also play a vital role in decreasing salt intake. A supportive social setting helps in improving the dietary habits of the people who eat foods with a higher salt concentration [62].

### 4.4. Perceptions on Dietary Salt Reduction

A study by Ma et al. (2014) indicated that very few people in the USA are aware of the dangers of taking high salt-content foods and display a lack of knowledge regarding the salt content in the packaged food products [63]. This issue can be resolved by implementing education-based intervention strategies such as community awareness campaigns, school-based education, and media campaign regarding salt reduction [57].

Furthermore, individual and organizational characteristics play a vital role in salt intake. Mallia et al. (2012) conducted a survey that showed that 90% of the respondents added salt to their dishes for improving the flavor and taste of the foods [49]. With regards to gender, many of the women add salt to the foods during cooking compared to the men who add more salt to their food on the table. Moreover, many women believed that salt added during cooking was the major source of salt in their diet. The study indicated that the Greek adults aged from 25 to 34, 35 to 44, and 45 to 54 years had a better knowledge regarding the harmful effects of salt on their health in comparison to respondents above 55 years of age [66]. With regards to individual knowledge, one-third of the Australian adults (32%) stated that they added salt to their plates on the table. Seventy three percent of the respondents stated that they had seen or heard the salt reduction messages [55]. Some of the factors which contributed to a higher salt intake included age and gender (which could not be modified), in addition to knowledge, attitude, and practices followed by the individuals that can be modified and improved by education [7,89,90].

Very few studies were conducted in the LMICs, particularly in the South-East Asian regions. The cost-effectiveness of implementing salt reduction strategies in these countries was not reported. Hence, a detailed and comprehensive study needs to be carried out in the future. A majority of the barriers, enablers, and perceptions of the individuals highlighted in this scoping review can help the governments and policymakers in the LMIC regions for developing, implementing or monitoring the strategies and policies for reducing the salt intake among the people.

## 5. Conclusions

This review defined the various barriers affecting the implementation of salt reduction strategies, which were (1) Lack of menu and food variabilities, (2) Lack of technical skills for cooking with lower salt, (3) Profit loss due to loss of sales, and (4) Lack of technical expertise for reformulation. The enablers themes are (1) Salt reduction intervention, (2) Food analysis studies and salt intake measurement, (3) Salt substitute accessibility, and (4) Gradual reduction of salt approach and Training availability. The perceptions themes are (1) Individual or organizational characteristics effects on salt intake, (2) Negative effects of high salt diet, and (3) Low salt food was recognized as tasteless. It was seen that the barriers, enablers, and perceptions regarding the dietary salt reduction in the out-of-home sectors were multifaceted. It is important to implement strategies, policies, and initiatives for addressing the barriers. For this purpose, a multisectoral collaboration from different stakeholders such as policymakers, food industries, catering operators, and food vendors is necessary to reduce the salt intake in the out-of-home sectors.

## Figures and Tables

**Figure 1 ijerph-18-08099-f001:**
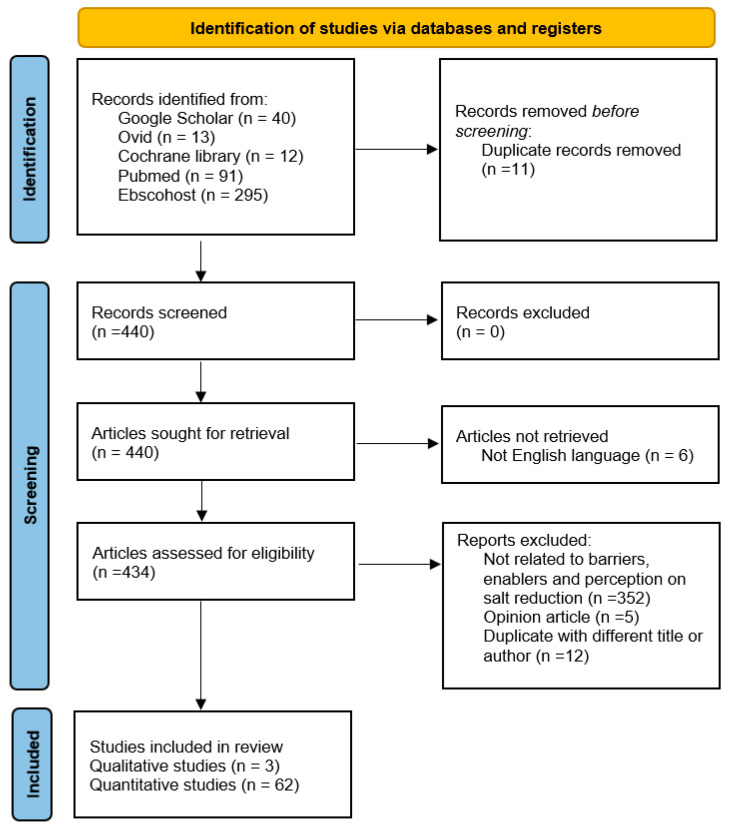
PRISMA-scoping review flow chart of included and excluded studies.

**Table 1 ijerph-18-08099-t001:** List of keywords and synonyms generated as search terms.

Barriers	Enablers	Perception	Salt Reduction	Out-of-Home
limitation	motivate	view	sodium reduction	street
difficulty	promote	attitude	salt reduction	cafeteria
restriction	help	mindset		work place
drawback	ease	willingness		food industries
	aid	readiness		
	policymakers			

**Table 2 ijerph-18-08099-t002:** Summary of studies’ characteristics.

No.	Research Method	Study Design	Country	Study Year	Study Population	References
1.	Qualitative	Focus group discussions and in-depth interviews	USA	2009	Policy makers (30 key informant interviews from the County of Los Angeles government departments)	[20]
			South Korea	2014	25 catering operators	[21]
			India	Not stated	The central and state governments (n = 4); the India Office and the South East Asia Regional Office of the World Health Organization (n = 5); the Indian Council of Medical Research (n = 2); the World Bank (n = 1); food manufacturers (n = 6); the academia (n = 10); the National Institute of Nutrition (NIN) (n = 3); the civil society (n = 4); community members (n = 7) from the urban and rural areas	[22]
2	Quantitative	Review	Global	2014–2017		[4,5,23,24,25]
			USA	2013		[26]
			Africa	2014–2015		[27,28]
			Europe	2015		[29]
			Australia	2015		[30]
			China	2018		[31]
			Eastern Mediterranean region	2020		[32]
		Intervention	Europe	2012	155 school children aged 10–12 years old) randomized equally to control and intervention groups	[33]
				2013	Intervention market survey (nine types of partially baked, reduced-salt breads) randomized equally	[34]
				2015–2016	371 employees from work organization aged 20 to 65 years (128 intervention, 13 control, and 230 non-participants)	[35]
			Brazil	2011	206 hypertensive patients aged 20 and above randomized into 101 control and 105 intervention participants [36]	[36,37]
			Canada	2011	987 with or without hypertension public randomized into four mock packages	[38]
			USA	2010 [39] 2014 [40] 2013–2015 [41]	Implementation of programme [39,40] 21 organizations including seven hospitals, eight YMCA branches, four community health centers, and two organization that served individuals experiencing homelessness [41]	[39,40,41]
				2019	Intervention of sodium reduction laws from all 50 states	[42]
			Australia	2011–2014	572 intervention participants aged 20 years and above	[43]
			Japan	2015	35 housewives aged 40 years and above and 33 family members aged 20 years and above randomized into 32 control and 36 intervention participants	[44]
			Korea	2019	Policy makers	[45]
		Retrospective	Australia	2008–2011 [46] 2018 [47]	107 ready meal products in 2008, 313 in 2009, 219 in 2010, and 265 in 2011 [46] Retrospectively applied Impact of Translational health research (FAIT) methods [47]	[46,47]
			USA	2009–2014	Retrospective assessed target achievement and change in sales-weighted mean sodium density in top-selling products in 61 food categories in 2009 (n = 6336), 2012 (n = 6898), and 2014 (n = 7396)	[48]
		Cross sectional	Europe	2009 [49] 2008 [50]	31 restaurant catering operators [49] 215 restaurants (140 and 75 from Wirral and Liverpool City Councils, respectively) [50]	[49,50]
				2014 [51,52] 2011–2015 [53]	6348 food products [51] 501 public consumers (aged 18 years and above) [52] 5759 prepacked foods [53]	[51,52,53]
			Australia	2010 [54] 2015 [55,56,57]	907 McDonald’s and Subway lunchtime customers (aged 16–84 years) [54] 338 participants aged 18–65 years [55] 2559 Victorian adults aged 18–65 years [56] 479 adults aged 18–64 years [57]	[54,55,56,57]
				2014	1153 soup products	[58]
			South Korea	2013 [59] 2014 [60]	257 females aged 25 to 49 years [59] 738 consumers aged 18 years and above	[59,60]
				2014 [61] 2018 [62]	104 food service personnel working at 17 worksite cafeterias aged 18 years and above [61] 312 restaurant managers and chefs [62]	[61,62]
			USA	2012–2013	221 Chinese take-out restaurants	[63]
				2013–2014 [64] 2011–2012 [65]	450 adults aged 18 years and above [64] 2948 children aged 2–18 years and 4878 adults aged > 18 years [65]	[64,65]
			Greece	2011	3609 Greek adults aged over 25 years	[66]
			Canada	2011 [67] 2014 [68]	1013 individuals aged 18 years and above [67] 27 healthcare professionals [68]	[67,68]
			Mongolia	2010	1027 residents aged 25 to 64 years	[69]
			Vietnam	2013–2014	513 participants aged 25 to 64 years	[70]
			Brazil	2011–2013	21 food categories	[71]
			Japan	2013	267 school adolescents aged 12 to 18 years	[72]
			Vanuatu	2016–2017	753 participants aged between 18 and 69 years from rural and urban communities	[73]
			Latin America China	2015–2016	This study examined sodium levels in 12 categories of packaged foods (n = 16, 357) sold in 14 Latin America countries	[74]
			Netherlands	2006–2015	The study population were, n = 317 (2006), n = 342 (2010), n = 289 (2015) in Netherlands	[75,76]
				2007–2010	Dutch National Food Consumption Survey	[76]
		Projection Modelling approach	China	2019 [75] 2016 [76]		[77,78]
			Netherlands	2016		[79,80,81]
			USA	2013		[82]

Note: USA: United States of America.

**Table 3 ijerph-18-08099-t003:** Summary of the papers related to barriers, enablers, and perceptions of dietary salt reduction in the out-of-home sectors.

	Study (Location)	Findings	References
Barriers			
Lack of menu and food variabilities	Gase et al. 2011 (USA)	Costs and unavailability of low-sodium foods	[20]
	Jaworowska et al. 2012 (UK)	Popular hot takeaway meals have alarming high salt content	[50]
	Lee et al. 2015 (Korea)	Need diverse menus at worksite	[60]
	Lee and Park 2016 (Korea)	Lack of various delicious low sodium menus	[61]
	Ahn et al. 2019 (Korea)	Maintaining taste and hindering the cooking process were the main barriers to reduce sodium use	[62]
	Dunford et al. 2020 (USA)	The majority of sodium sources available in the market are from packaged and processed food	[65]
Lack of technical expertise for reformulation	Gase et al. 2011 (USA)	Lack of knowledge and experience in operationalizing sodium standards	[20]
	Maalouf et al. 2013 (USA)	No comprehensive sodium content data in restaurant foods	[26]
Lack of technical skills for cooking	Gase et al. 2011 (USA)	Features of food service settings including their existing food standards, other nutritional mandates, the populations they serve, and current contracts	[20]
	Kloss et al. 2015 (Europe)	Food safety issues, consumer acceptance concerns, cost concerns and complications arising from the use of sodium alternatives	[29]
	Lee and Park 2016 (Korea)	Limited methods of sodium-reduced cooking in worksite cafeteria	[61]
	Grime et al. 2017 (Australia)	No low salt foods available—eating out	[56]
Profit loss due to salt reduced foods	Park et al. 2016 (Korea)	Environmental factor such as pressure to maintain profit margins	[21]
	Gupta et al. 2018 (India)	Decreased sales due to salt reduction	[22]
	Dodd et al. 2019	Economic assessment of salt reduction efforts	[47]
Others	Gase et al. 2011 (USA)	Complexity of food service arrangements, lack of consumer demand for low-sodium foods, undesirable taste of low-sodium foods, preference for prepackaged products, and existing multiyear contracts that are difficult to change	[20]
	Christoforou et al. 2013 (Australia)	Failure of voluntary industry efforts suggest a regulated approach for ready meal products	[46]
	Kloss et al. 2015 (Europe)	Limited motivation among food manufacturers	[29]
	Curtis et al. 2016 (USA)	Industry slow progress	[48]
	Inguglia et al. 2017	Microbial safety in low sodium meat products	[25]
	Gupta et al. 2018 (India)	Social and cultural beliefs, a large unorganized food retail sector, and the lack of proper implementation of even existing food policy	[22]
	Ahn et al. (2019) Korean	Purchasing experience after comparing sodium content in the nutritional labeling	[62]
Enablers			
Training	Ma et al. 2014 (USA)	Training in food preparations, procurement and marketing	[63]
	Park et al. 2016 (Korea)	Skills and techniques related to measuring sodium content and preparing reduced sodium meal (RSM) were emphasized by the interviewees	[21]
	Land et al. 2016 (Australia)	Good knowledge	[43]
Salt substitute	Cotter et al. 2013 (Portugal)	Using herbs and spices	[33]
	Land et al. 2016 (Australia)	Using substitutes	[43]
	Quilez et al. 2016 (Spain)	Replacing salt with potassium citrate in bread	[34]
	Van Buren et al. 2016 (Netherlands)	Replacement of sodium chloride by potassium chloride, particularly in key contributing product groups	[79]
	Johnson et al. 2017 (Global)	Use of low sodium salt substitutes	[24]
	Brooks et al. 2017 (USA)	Increase availability of lower-sodium products	[41]
	Lacey et al. (2018) (Ontario)	Lower-sodium foods, increased availability of pre-packaged lower-sodium products	[68]
Gradual reduction of salt	Coxson et al. 2013 (USA)	A gradual reduction in dietary sodium	[82]
	Losby et al. 2014 (USA)	Gradual and voluntary reduction of sodium content	[40]
	Curtis et al. 2016 (USA)	National target setting (voluntary target)	[48]
	Johnson et al. 2017 (Global)	Specific sodium target	[24]
	Levi et al. 2018 (Australia)	A 6% reduction in sodium levels in soups overall was found from 2011 to 2014	[58]
	Yang et al. 2019 (China)	Stages of salt reduction in food industry	[77]
	Park et al. 2020 (Korea)	South Korea implemented its National Plan to Reduce Sodium Intake, with a goal of reducing population sodium consumption by 20%, to 3900 mg/day, by 2020	[45]
Intervention	Christoforou et al. 2013 (Australia)	Reformulation	[46]
	Coxson et al. 2013 (USA)	Range of proposed public health intervention	[82]
	Cotter et al. 2013 (Portugal)	Weekly lessons on danger of high salt intake, (theory), practical lessons in school garden of planting and collection of herbs for salt substitution at home	[33]
	Lima et al. 2013 (Brazil)	Feasible dietary approach	[36]
	Wong et al. 2013 (Canada)	Sodium claim Active education program of sodium reduced cooking	[38]
	Losby et al. 2014 (USA)	Product substitutions, recipe modifications, and cooking from scratch	[40]
	Johnston et al. 2014 (USA)	A social marketing approach was used to educate consumers about the hidden sources of dietary sodium	[39]
	Webster et al. 2014 (Global)	Legislation	[23]
	Kim et al. 2014 (Korea)	Labels influenced consumer satisfaction	[59]
	Kloss et al. 2015 (Europe)	Food reformulation efforts have been made in the bread, meat, dairy, and convenience foods industries	[29]
	Trieu et al. 2015 (Global)	Reformulation with sodium content targets Consumer education Front-of-pack labelling Taxation of high-salt foods Public institution	[4]
	Webster et al. 2015 (Australia)	Drop Salt Campaign by NGOs and food industry to advocate the government to develop a national strategy to reduce salt Food reformulation Health Star rating front pack labelling	[30]
	Enkhtungalag et al. 2015 (Mongolia)	Pinch Salt intervention to reduce salt consumption among factory workers	[69]
	Do et al. 2016 (Vietnam)	COMBI intervention—effective in lowering salt intake and improving knowledge and behaviours	[70]
	McLaren et al. 2016 (Global)	Population level intervention	[5]
	Wang et al. 2016 (China)	Population wide dietary salt reduction policies	[78]
	Takada et al. 2016 (Japan)	Measure the difference in estimated daily salt intake by spot urine sampling of housewives and their family members 2 months after intervention between the groups	[44]
	Land et al. 2016 (Australia)	Read labels	[43]
	Okuda et al. 2017 (Japan)	Home environment and salt-use behaviour intervention in secondary school	[72]
	Brooks et al. 2017 (USA)	Increase availability of lower-sodium products	[41]
	Johnson et al. 2017 (Global)	Behavior change intervention	[24]
	Zhang et al. 2018 (China)	AIS—Application Based Intervention Study using mobile application to reinforce and maintain lower salt intake RIS—Restaurant based Intervention Study for consumers, cooks, and restaurant managers HIS—Housewife Intervention Study for family chef CIS—Comprehensive Intervention Study for evaluating all interventions	[31]
	Gupta et al. 2018 (India)	The development and adoption of the National Multi-Sectoral Action Plan	[22]
	Trieu et al. 2018 (Australia)	Components chosen in an intervention is important	[57]
	Yang et al. 2019 (China)	Vigorous advancements of salt reduction actions	[77]
	Sloan et al. 2020 (USA)	Laws: Labels, workplace, vending machines	[42]
	Doggui et al. 2020 (Eastern Mediterranean Region)	Mandatory regulatory measures for universal salt iodization	[32]
Food analysis studies and salt intake measurement	Cotter et al. 2013 (Portugal)	24 h urinary sodium excretion analysis	[33]
	Antoniolli et al. 2014 (Australia)	Sodium and saturated fat contents were calculated from company websites	[54]
	Losby et al. 2014 (USA)	Sodium nutrient analysis	[40]
	Korosec et al. 2014 (Slovenia)	Market leaders have lower salt content through comparison of the category average sodium content of prepackaged foods	[51]
	Enkhtungalag et al. 2015 (Mongolia)	Salt in tea contribute 30% of daily salt intake	[69]
	Johnson et al. 2017	Spot or 24 h urinary sodium salt intake measurement	[24]
	Nilson et al. 2017 (Brazil)	Monitoring sodium content of food	[71]
	Pravst et al. 2017 (Slovenia)	Sales weighted sodium content	[53]
	Arcand et al. 2019 (Latin America)	Sodium content in packed foods and positive impact of menu labeling	[74]
	Temme et al. (2017)	The salt content of bread, certain sauces, soups, potato crisps, processed legumes and vegetables have been reduced over the period 2011–2016 in Netherlands. However, median salt intake in 2006 and 2015 remained well above the recommended intake of 6 g	[75]
Others	Cotter et al. 2013 (Portugal)	Blood pressure measurement	[33]
	Antoniolli et al. 2014 (Australia)	Nutritionally promoted fast foods may contain less sodium when selected	[54]
	Ma et al. 2014 (USA)	Customer demand maintained strategies and support	[63]
	Losby et al. 2014 (USA)	Identifying partners with shared experience and common goals Working towards sustainable sodium reduction	[40]
	Sookram et al. 2015 (Africa)	Overview of WHO supported interventions on salt intake reduction among Member States of the African	[28]
	McLaren et al. 2016	Multicomponent intervention incorporating product reformulation among men	[5]
	Curtis et al. 2016 (USA)	Monitoring through partnership of local as well as state and national health organizations	[48]
	Park et al. 2016 (Korea)	Key stakeholders’ psychosocial characteristics and environment factors	[21]
	Inguglia et al. 2017 (Global)	High pressure processing and power ultrasound, seem to be promising to ensure microbiological safety in low-sodium meat products	[25]
	Lacey et al. 2018 (Ontario)	Group purchasing organizations, government prioritizing, and providing support and resources. Improved tastes of lower-sodium foods	[68]
	Gupta et al. 2018 (India)	Most of the stakeholders were in alignment with the need for a salt reduction programme in India to prevent and control hypertension and related cardiovascular diseases	[22]
	Ahn et al. 2019 (Korea)	Supportive social environment, improving dietary habits of eating high salt foods	[62]
	Hendriksen et al. (2015a)	Predictive study using population health modelling tool showed that reduction of salt intake to 5 g per day is expected to substantially reduce the burden of cardiovascular disease and mortality in several European countries	[81]
	Hendriksen et al. (2015b)	Modification of food composition or by alteration of behaviour may substantially reduce the median sodium intake using two scenarios from the National Food Consumption Survey 2011.	[76]
	Hendriksen et al. (2017)	Different health impact model assessment from seven population health impact models may affect the health impact estimate, however, the estimated impact of salt reduction was substantial in all of the models used, emphasizing the need for public health actions	[80]
Perceptions			
High salt is bad for health	Mallia et al. 2012 (Europe)	99% of the respondents were aware which foods are low or high in salt	[49]
	Kim et al. 2014 (Korea)	Consumers’ knowledge of the relationship between diets high in sodium and an increased risk of developing previously reported sodium-related diseases	[59]
	Mezue et al. 2014 (Nigeria)	Plan for a population-wide salt reduction strategy	[27]
	Ma et al. 2014 (USA)	Lack of knowledge on the danger of high salt intake	[63]
	Enkhtungalag et al. 2015 (Mongolia)	Mongolia has one of the highest rates of stroke, high salt intake from tea (30%)	[69]
	Trieu et al. 2018 (Australia)	The proportion who understood the adverse effects of salt (+9.0%, *p* = 0.049)	[57]
	Sparks et al. 2019 (Vanuatu)	Total of 83% of participants agreed that too much salt could cause health problems	[73]
	Ahn et al. 2019 (Korea)	Knowledge edge of the recommendation of salt, difference between sodium and salt	[62]
Low salt food was recognized as tasteless	Gupta et al. 2018 (India)	Participant perceived that reduced salt will make food not tasty	[22]
	Lacey et al. 2018 (Ontario)	37% believe that the patient would decrease with sodium reduction	[68]
Individual or organizational characteristics	Mallia et al. 2012 (Europe)	90% of the participants added salt to dishes to enhance flavour and improve taste	[49]
	Kim et al. 2014 (Korea)	Current consumer knowledge on the sodium content in food products was high	[59]
	Losby et al. 2014 (USA)	Understanding the complexity of the meal’s system for older adults	[40]
	Marakis et al. 2014 (Greece)	Gender (more women added salt during cooking, less on the plate compared to men, more women believed that salt added during cooking was the main source of salt in the diet). For age, participants aged 25–34, 35–44, and 45–54 years old had better knowledge of the harmful effects of salt on health compared with 55 years and above	[66]
	Enkhtungalag et al. 2015 (Mongolia)	Most participants knew that salt was bad for health, few were taking efforts to reduce intake, and many were consuming highly salty meals and tea	[69]
	Regan et al. 2016 (Ireland)	A series of multiple regressions revealed that individual attitudes and beliefs related to health and salt were stronger predictors of support than sociodemographic factors, lifestyle or knowledge	[52]
	Quilez et al. 2016 (Spain)	Acceptance of the reduced salt bread	[34]
	Lee and Park 2016 (Korea)	Most of the participants had relatively high levels of perception regarding the importance of sodium reduction	[61]
	Grimes et al. 2015 (Australia)	83% believed that Australians eat too much salt	[56]
	Harnack et al. 2017 (USA)	Sodium added to food outside the home accounted for ≈70% of dietary sodium intake	[64]
	Trieu et al. 2018 (Australia)	A total of 73% reported that they had heard or seen the salt reduction messages	[57]
	Sparks et al. 2019 (Vanuatu)	A total of 83% of participants agreed that too much salt could cause health problems	[73]
	Ahn et al. 2019 (Korea)	The majority (82%) was willing to reduce sodium in restaurant foods under the support of local government and they desired the promotion of participating restaurants and education on cooking skills to reduce sodium	[62]

Note: UK: United Kingdom; USA: United States of America.

## Data Availability

All data are contained within this manuscript.

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
