# Peer review of "Barriers, Enablers, and Perceptions on Dietary Salt Reduction in the Out-of-Home Sectors: A Scoping Review"

_ijerph, 2021, doi:10.3390/ijerph18158099_

Round 1

Reviewer 1 Report

My comments are attached as a separate file.

Author Response

Dear reviewer, 

Thank you very much for your review, we appreciate all your valuable comments, we hope that we have addressed all your concerns, please find the attached file. 

Thank you very much. 

Reviewer 2 Report

Dear colleagues,

This is important topic.

Submitted review is conducted using common approach and it is very comprehensive.

My suggestion is to go over comments and to consider slight modifications for better flow.

I would like to see more details in the flowchart regarding category “not related to topic’.

In addition, if it is possible, I would like to see the list with the original search terms.

Thank you.

All the best!

Author Response

Dear reviewer, 

Thank you for your review, we appreciate all your valuable comments. I hope we have addressed all your concerns, please find the attached file for our corrections. 
